# Trends in Incidence and Case Fatality Rates of Heart Disease and Its Subtypes in Korea, 2006–2015

**DOI:** 10.3390/ijerph17228451

**Published:** 2020-11-15

**Authors:** Su Ra Seo, Shin Yi Jang, Sung-il Cho

**Affiliations:** 1Graduate School of Public Health, Seoul National University, Seoul 08826, Korea; scho@snu.ac.kr; 2Health Insurance Research Institute, National Health Insurance Service, Wonju-si 26464, Korea; 3Imaging Center, Heart Vascular Stroke Institute, Samsung Medical Center, Seoul 06351, Korea; Bautai@hanmail.net

**Keywords:** heart disease, subtypes of heart disease, incidence, fatality rate, trends

## Abstract

Heart disease (HD) is the second leading cause of death in Korea. Several studies in Korea have analyzed the trends of incidence and mortality of myocardial infarction (MI) and ischemic heart disease (IHD), but few have investigated incidence and mortality trends of HD and its subtypes. The aim of this study was to assess the national trends in incidence and case fatality rates of overall HD and its subtypes (including IHD, heart failure (HF), arrhythmia, hypertensive HD (HHD), valvular HD, pulmonary HD, and others) in Korea between 2006 and 2015. Using records from the National Health Insurance Service (NHIS) claims database (2003–2015) and by obtaining the causes of death (Korean Statistical Information Service, 2006–2017), we analyzed the crude and age-standardized incidence rates from 2006 to 2015 and the case fatality rates from 2006 to 2017 of HD and its subtypes. Between 2006 and 2015, the incidence of overall HD changed minimally, but the age-standardized incidence of HD decreased from 210.0 persons per 100,000 populations in 2006 to 161.3 persons in 2015. However, incidence rates have increased in arrhythmia, HD other, pulmonary HD, and the case fatality rates have increased in HF, valvular HD, and HD other. Therefore, it is essential to continuously monitor the incidence and case fatality rates of HD and its subtypes and expand the focus onto prevention and treatment strategies from MI or IHD to various HD subtypes. Active prevention and management are needed to alleviate the burden of HD due to an aging population in Korea.

## 1. Introduction

The mortality rate of coronary heart disease (CHD) increased in the early 20th century, peaked in the late 60s, and began to decrease thereafter in Western countries. The mortality rates of CHD have decreased in the US, UK, and the central and eastern European countries in the last 30 years [1,2,3]. However, the countries in the Asia-Pacific region such as Korea, Japan, Singapore, and China have revealed a different trend. The reason may include rapid urbanization over the past few decades [4,5,6,7]. Several studies in the US have analyzed the trends in the mortality rates of heart disease (HD) and its subtypes [8,9], but few have investigated incidence trends. Heart disease (HD) is the second commonest cause of death in Korea [10]. Although studies have assessed the trends of the incidence of acute myocardial infarction (AMI) and ischemic heart disease (IHD) [11,12,13,14], details about incidence trends of HD and its subtypes is scarce in Korea.

HD encompasses a wide range of heart conditions. HD includes various subtypes, comprising ischemic heart disease (IHD), heart failure (HF), hypertensive heart disease (HHD), valvular heart disease, arrhythmia, pulmonary HD, and other HD [1,8,9]. For effective strategies to improve the health status, to encourage healthy aging, to reduce years lived with disability, and to decrease the economic burden of disease of HD, it is important to clearly identify the current status and characteristics of the disease and its dynamics. There are various measurements of the effects of a disease in the population, including incidence, prevalence, complications, mortality, risk factors, quality of life, health care use, and economic costs, etc. [1]. Incidence and fatality rates are the most important estimates for timely identification of changes in unfavorable factors, and they help provide important feedback to alter health policy priorities for achieving disease control and decreasing its burden [2]. However, there is a lack of national representative data on the incidence of HD as well as its subtypes. Several studies have inspected the trends of AMI and IHD incidence and mortality in Korea [11,12,13,14]. Despite considerable information on IHD incidence and mortality trends, less is known about trends for overall HD and its other subtypes. Thus, we examined the national trends in incidence and case fatality rates of overall HD and its subtypes in Korea between 2006 and 2015.

## 2. Materials and Methods

### Ethics Approval and Consent to Participate

This study and the use of NHIS data were approved by the Institutional Review Boards of Seoul National University (SNU IRB No. E1901/001-001) and National Health Insurance Service (NHIS-2019-1-200), respectively. The study used secondary data from NHIS (anonymized data), and, therefore, the requirement for patient consent was waived. The investigations were carried out in accordance with the Declaration of Helsinki.

We followed the subsequent steps for calculating the yearly HD incidence. First, we collected data about inpatients with HD as the primary diagnosis between 2003 and 2015 from the National Health Insurance Service (NHIS) claims database. These data included the total medical utilization of about 52 million Korean populations (Appendix A). In addition, people who were not inpatients but had died from HD during each year were added based on death records from the Korean Statistical Information Services to calculate incidence. The International Classification of Disease, 10th Revision (ICD-10) classification was used to define HD, and the comprehensive index for HD was determined as per ICD-10 codes I00–I09, I11, I13, and I20-I51 [1,8,9].

We excluded the person counts of HD inpatients prior to the target year to delineate the incidence for each year. The incidence of HD and its other subtypes, until 2015, were also analyzed in the same way.

Third, we calculated the crude incidence rate and categorized it based on gender and different age groups. The total population was used as the denominator for calculating the incidence for each year [15]. The age groups were subdivided into those less than 40 years old, 40–49 years old, 50–59 years old, 60–69 years old, 70–79 years old, and 80 years or older.

Fourth, the age-standardized incidence was determined from the World Standard Population constructed by WHO for the period 2000–2025 based on the yearly modifications in the ages of the population [16]. The method for calculating the age-standardized incidence of HD subtypes was the same as above. The HD subtypes were classified as IHD (I20–I25), HF (I50), arrhythmia (I47–49), HHD (I11, I13), valvular HD (I34–I38), pulmonary HD (I26–I28), and others (I00–I09, I30–I33, I40–I46, I51) [1,8,9]. Incidence rates for overall HD and the sum of incidence rates for the various HD subtypes could be different. For example, a person who is admitted for angina pectoris (AP) in 2006 and is also an inpatient for HHD in 2011 is considered a new occurrence of HD in 2006 but not in 2011, because HD includes AP and HHD. However, when assessed separately the event is counted as a new occurrence of AP in 2006 and new occurrence of HHD in 2011.

We assessed the case fatality rates, from the cause and date of death data obtained from the Korean Statistical Information Service by linking the personal identification number (by using a non-identified number for analysis) with the incidence of HD in the year. Only those who died from HD were included, while those who died from other causes of death were excluded. We used the NHIS claims database to calculate the dates between the date of first admission for HD and the date of death due to HD in order to evaluate the case fatality rate, which was then classified into a 7-day, 30-day, 1-year, 3-year, and 5-year cumulative case fatality rate [1,12,17]. The case fatality rates were calculated using the number of deaths from HD in a designated period per the number of new inpatient cases. Since the death data were considered until 2017, the 3-year cumulative case fatality rates were not calculated for 2015 and the 5-year cumulative case fatality rates 2013–2015, respectively, to avoid underestimation.

The NHIS claims database needed addition annual information to construct data and improve its stability and integrity. The aim of this study was to assess trends in case fatality rates too, which required a supplementary observation period to calculate the 1-year cumulative case fatality rates. Thus, we decided to use the NHIS claims database for analyzing the trends in incidence between 2003 and 2015 and the cause of death from the Korean Statistical Information Service for trends in case fatality between 2006 and 2017.

Statistical Analysis System (SAS) 9.4 (SAS Institute, Cary, NC, USA) was used for the analysis of the incidence, incidence trend (Cochran-Armitage trend test), and case fatality rates of HD in the Korean population.

## 3. Results

### 3.1. Incidence Rates of HD

Incidence and crude rates of HD in the total (*p* for trend < 0.0001) and male population (*p* for trend < 0.0001) increased over the past 10 years and decreased in females (*p* for trend < 0.0001) (Table 1, Appendix A). The age-standardized incidence rates of HD decreased in total and in both males and females. The mean age of occurrence increased from 63.6 years in 2006 to 65.1 years in 2015. It had increased 1.8 years within 10 years. The mean age of occurrence in males and females increased from 59.9 and 66.8 years in 2006 to 61.9 and 69.2 years in 2015, respectively. The mean age of occurrence of HD in females was approximately 7 years higher than in males. This difference in the mean age has changed little in 10 years.

### 3.2. Incidence of HD Subtypes

The annual incidence of HD subtypes, and the crude rate of IHD decreased slightly from 162.6 persons per 100,000 population in 2006 to 158.2 in 2015; however, the age-standardized rate of IHD was 143.9 persons in 2006 per 100,000 population to 103.5 persons in 2015. The difference in the crude rates was small, but the age-standardized rates revealed a relatively larger decrease in 10 years. (Table 2, Appendix A). The crude and age-standardized incidence rates of arrhythmia, HD other, and pulmonary HD increased, but HHD decreased for 10 years (Table 2, Appendix A).

The incidence trends of arrhythmia, HF, HD other, pulmonary HD, and valvular HD were increased and HHD was decreased significantly. However, the incidence trend of IHD was not significant (*p* for trend = 0.1178) (Table 2).

Depending on the subtypes of HD, the mean age of occurrence revealed differences. The mean age of occurrence of HF and HHD were in the mid-70s between 2006 and 2015. The mean age of occurrence of the other HD subtypes including IHD, arrhythmia, pulmonary HD, valvular HD was mid-60 s. Hence, it showed that the mean age of occurrence of HF and HHD was 10 years more than the other subtypes of HD and mean age of occurrence of overall HD and its subtypes had increased for 10 years (Table 2, Appendix A).

### 3.3. Case Fatality Rates of HD and Its Subtypes

The case fatality rate of HD for 7 days increased from 1.2% in 2006 to 1.4% in 2015; and the cumulative case fatality rate of HD for 30 days increased from 2.2% in 2006 to 2.5% in 2015. However, cumulative case fatality rates for 1, 3, and 5 years had decreased. Case fatality rates of IHD and pulmonary HD were found to decrease during the 10 years (Figure 1, Appendix A). In contrast, the case fatality rates of HF, valvular HD, and HD other were relatively higher than the other subtypes of HD and had increased during the 10-year period. In particular, the case fatality rate of HF for 7 days was 3.3%; the cumulative case fatality rate for 30 days was 6.7%; for 1 year, it was 13.0% in 2015; for 3 years, it was 19.1% in 2014; and for 5 years, it was 22.3% in 2012. The cumulative 5-year case fatality confirmed that one-fourth of all cases of heart failure died within 5 years due to heart diseases. Cumulative 5-year case fatality rates for valvular HD, HHD, and HD other were approximately 15% and for pulmonary HD was approximately 10% in 2015. It should be noted that the case fatality rates of overall HD, IHD, and pulmonary HD had a tendency to decrease between 2006 and 2017, but the case fatality rates had increased and were higher in HF, valvular HD, and HD other compared to the other subtypes of HD. The case fatality rates by subtypes of HD in females were higher than in males. This difference of case fatality rates by sex has changed little in 10 years (Appendix A).

## 4. Discussion

In Korea, HD carries the second highest death rate (62.4 per 100,000 in 2018) following cancer, and the recent trends reveal an increase [10]. There are a variety of ways to measure the population impact of a disease, and one of the most critical measurements is the incidence rate of a disease [11]. However, there is a lack of information on the incidence rates of HD and its subtypes rather than the mortality rates. This is because it was difficult to detect the incidence of HD than to assess mortality due to the easy availability of representative data about the cause of death from the National Statistical Office. The cost for investigating the incidence is high in the community cohort, and it is not easy to obtain representative data [3]. Additionally, the incidence had to be calculated from various sources (outpatient, inpatient, died out of hospital), which made it challenging [11]. This study showed that the crude incidence rate of overall HD, arrhythmia, HF, HD other, pulmonary HD, valvular HD excluding IHD, and HHD had increased between 2006 and 2015. However, the age-standardized incidence rates of HD and its subtypes had decreased except for arrhythmia, HD other, and pulmonary HD. During the 10-year period, the decrease in the age-standardized incidence rates of HD was attributable to the reduction in IHD, HF, and HHD incidence. Furthermore, the growth of the aging population was one of reasons for increasing crude rate incidence [10,11,12].

Major risk factors for cardiovascular disease referred demographic factor (age, sex), health habits (smoking), and prevalence of metabolic disorders (obesity, hypertension, diabetes, hypercholesterolemia) frequently [18,19,20]. However, obesity, hypertension, diabetes, and hypercholesterolemia except smoking had changed a little. Between 2001 and 2014, the age-adjusted smoking prevalence decrease significantly, from 64.0% to 47.1% (*p* = 0.03), but decreasing prevalence of hypertension (from 27.9% to 25.6%) and increasing prevalence of diabetes (from 7.3% to 9.9%) and hypercholesterolemia (from 8.6% to 12.6%) were not significant among men in Korean representative survey (Korean National Health Nutrition Examination Survey: KNHANES) [19]. The other study showed similarly the prevalence of diabetes, hypertension, obesity, and hypercholesterolemia except smoking rate changed a little from 2008 to 2013 [20]. These trends of risk factors could not explain enough decreasing pattern of age-standardized incidence patterns for overall HD, IHD, HF, and HHD during the 10 years in this study. The increasing of healthy elderly assumed that increasing patterns of crude incidence rate and decreasing patterns of age-standardized incidence rates in overall HD and HF.

We found rising patterns of case fatality in HF, HD other, and valvular disease but not IHD and pulmonary HD. Cumulative 5-year case fatality rates were over 20% in HF and HD other, and over 15% in valvular disease and HHD. Although trends of incidence, mortality, and case fatality rates have been reported about AMI and IHD in Korea, to the best of our knowledge, our study is the first to report the incidence of HD and its subtypes. It included data about those who had died from HD out of hospital using the population survey between 2002 and 2015. We also calculated the cumulative 5-year case fatality rate for HD and its subtypes using data gathered from 2006 to 2017 in Korea.

There are three studies in Korea about the trends of AMI incidence that used different ways to identify inpatients based on the diagnostic code and time periods, making it difficult to compare them. One study examined the AMI incidence trends using the primary and secondary diagnoses of inpatients between 1997 and 2007 and revealed that the age-standardized incidence rates had increased from 50.4 patients per 100,000 population in 1997 to 78.3 patients in 2007 [11]. The other study conducted using the primary inpatient diagnosis between 2006 and 2010 revealed that the age-standardized incidence rates decreased from 45.7 patients per 100,000 population in 2006 to 29.4 patients in 2010 [12]. The third study between 2007 and 2016 included the primary diagnosis, duration of admission, type of tests performed, treatment given, and medication administered and reported that the age-standardized incidence rates decreased from 53.6 patients per 100,000 population in 2007 to 38.9 patients in 2011, followed by an increase from 39.5 patients in 2012 to 43.2 patients in 2016 [13]. However, it is not appropriate to compare the trends in these results with ours because the target disease, diagnosis criteria, and criteria for calculating the incidence were different (as they did not include HD deaths that occurred out of hospital).

Our analysis included IHD, not AMI alone (ICD-10 code of IHD: I20~I25; ICD-10 code of AMI: I21). IHD has three subtypes comprising angina pectoris (ICD-10 code: I20), MI (ICD-10 code: I21~I22), and IHD other (ICD-10 code: I23~I25). There was one study on the IHD mortality trends that showed a steady increase in mortality since 1985 until 2009 [17]. However, they calculated the age-adjusted mortality rate based on the age-groups, and hence, it could not be compared with our case fatality rates. There was another study that calculated the 7-, 30-, 90-day, 1-, and 3-year cumulative AMI case fatality trends between 2007 and 2016. A decrease for 10 years was noted, but it was for all-cause fatality not death according to HD subtypes, thereby preventing comparison with our results.

Korea has established “he first comprehensive plan for cardiovascular disease” in 2018 and is striving for cardiovascular disease management in all direction. The main strategies for implementation include improving public awareness and practicing healthy life; strengthening the management of high-risk groups and preceding disease; strengthening emergency response and treatment capabilities in the local community; establishing a patient persistence management system; and strengthening infrastructure, research, and research capability. In order to grasp the current status of cardiovascular disease and to evaluate policy effect, it is necessary to establish a representative monitoring system for the occurrence of cardiovascular disease, risk factor, and cause of disease. In addition, monitoring with same way continuously is important.

### 4.1. Advantages of Our Study

Our study had certain strengths. First, we provided an overview about the trends of incidence and case fatality rates from 7 days to 5 years for overall HD and its subtypes between 2006 and 2015 for the entire population of Korea. Second, although NHIS claims data were easily obtained, it was considered controversial due to its dubious diagnostic accuracy. However, several studies in Korea have validated the overall diagnostic accuracy as 82.0% [21], with a 71.4~96.8% accuracy for AMI [21,22] and 83.0% for cerebrovascular diseases (CVD) [23], and hence, it was not a significant limitation of this study. Third, we calculated incidence rates that included both inpatients and out-of-hospital deaths due to HD. It possibly led to improved accuracy in calculating incidence rates. Fourth, we calculated case fatality rates using only HD as the cause of death and not all-cause deaths. The overall accuracy rate of statistics on the causes of death was 91.9% [24]. Therefore, this may have been a more exact method for calculating case fatality rates for HD.

### 4.2. Limitations of This Study

This study had several limitations. First, this study did not include out-patients and no other diagnosis except the primary diagnosis (e.g., secondary diagnosis) for inpatients with HD and its subtypes, so we could have underestimated the incidence. The diagnostic accuracy of out-patients was 44.5% and that of secondary diagnosis of inpatients was 56.4% [21]. This is why we excluded the abovementioned patient cohorts.

Second, we had wash-out periods for only 3 years (2003–2005), and hence, it is possible that we underestimated or overestimated the incidence and incited ambiguous incidence and case fatality trends in the early period. For example, if an inpatient with HD in 2002 before the wash-out period was readmitted with recurrence of HD in 2006, then the case would be designated as a new case in 2006.

Third, we did not analyze the IHD subtypes, angina pectoris (AP), and myocardial infarction (MI). Generally, MI has higher fatality and severity than AP but the number of patients with AP are more than that of MI. Thus, this could have influenced the trends warranting attention to interpretation of results.

Fourth, we did not examine the risk factors that caused an increase or decrease in the incidence or case fatality trends of HD and its subtypes for 10 years. The aims of this study were to provide an overview about incidence and case fatality patterns for 10 years of HD and its subtypes, so further studies are required for analysis of risk factors.

## 5. Conclusions

Between 2006 and 2015, the crude incidence rates of overall HD have changed little, but the age-standardized incidence of HD has markedly declined. The reduction in IHD rates have contributed to this. However, incidence rates have increased in arrhythmia, HD other, and pulmonary HD, while the case fatality rates have been increased in HF, valvular HD, and HD other. Therefore, it is necessary to monitor incidence and case fatality rates of HD and its subtypes for optimizing public health priorities and policy to reduce the burden of HD in Korea.

## Figures and Tables

**Figure 1 ijerph-17-08451-f001:**
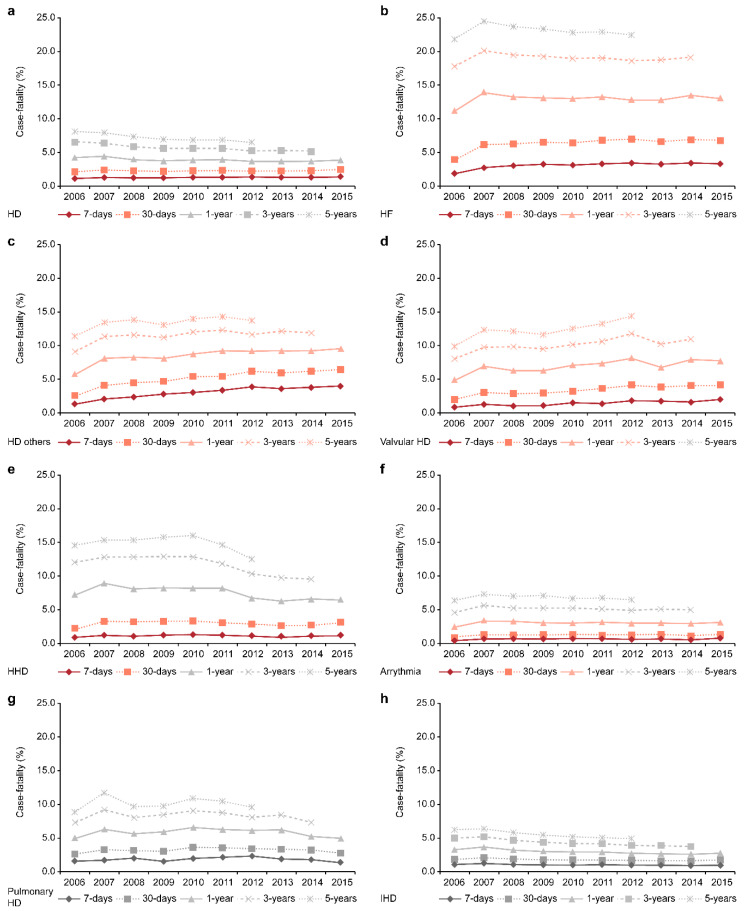
Trend in case fatality rates of HD subtypes in Korea, 2006–2015. (**a**) Overall of heart disease (ICD-10th code: I00–I09, I11, I13, I20–I51); (**b**) heart failure (ICD-10th code: I50); (**c**) other subtypes of HD (ICD-10th code: I00–I09, I30–I33, I40–I46, I51); (**d**) valvular HD (ICD-10th code: I34–I38); (**e**) hypertensive HD (ICD-10th code: I11, I13), (**f**) arrythmia (ICD-10th code: I47–I49); (**g**) pulmonary HD (ICD-10th code: I26–I28); (**h**) ischemic HD (ICD-10th code: I20–I25).

**Table 1 ijerph-17-08451-t001:** Annual numbers and incidence rates of HD in Korea, 2006–2015.

Variables	2006	2007	2008	2009	2010	2011	2012	2013	2014	2015
Total of HD										
Populations	49,238,227	49,672,388	50,001,057	50,290,771	50,581,191	50,908,645	51,169,141	51,448,491	51,757,146	52,034,424
Incidence numbers	115,127	117,758	115,801	116,441	119,244	123,168	126,105	122,821	127,636	125,388
Mean age, year	63.3	63.6	63.8	63.8	63.9	64.1	64.3	64.3	64.6	65.1
Age group, %										
0–39	5.1	5.2	5.1	5.1	5.0	5.0	4.8	4.9	4.6	4.5
40–49	11.1	10.8	10.4	10.2	10.0	9.5	9.3	9.4	9.2	8.7
50–59	20.4	20.0	19.8	20.4	20.7	21.4	21.2	21.4	21.3	20.6
60–69	27.5	27.0	26.8	26.1	25.9	24.8	24.7	23.9	24.1	24.4
70–79	24.6	25.0	25.4	25.4	25.5	25.6	26.1	26.0	25.5	25.3
80+	11.4	12.1	12.5	12.8	12.9	13.7	13.9	14.5	15.3	16.5
Total incidence rate (per 100,000 persons-year)										
Crude rate	233.8	237.1	231.6	231.5	235.7	241.9	246.4	238.7	246.6	241.0
Age-standardized rate	210.0	205.6	194.0	187.2	184.5	183.7	180.6	170.2	170.5	161.3
Age group, year										
0–39	20.4	21.4	21.2	21.6	22.1	23.0	23.0	23.3	23.2	22.3
40–49	150.4	147.5	138.1	134.6	135.2	131.9	132.7	128.6	130.6	122.3
50–59	421.3	399.6	369.4	361.4	351.4	351.7	343.6	326.5	328.4	308.4
60–69	846.7	821.3	772.2	746.3	741.2	728.9	721.7	657.3	652.9	601.4
70–79	1365.5	1331.9	1257.7	1191.5	1165.6	1143.9	1114.6	1044.2	1041.0	1002.6
80+	1852.8	1877.0	1777.4	1690.3	1639.0	1682.1	1622.5	1531.3	1548.9	1509.9
Male										
Populations	24,732,975	24,944,131	25,102,682	25,241,212	25,379,265	25,536,889	25,647,133	25,780,143	25,934,973	26,065,615
Incidence numbers	58,631	59,820	59,826	61,612	64,406	66,760	69,293	68,315	71,067	70,725
Mean age, year	59.9	60.2	60.4	60.5	60.8	61.0	61.2	61.2	61.5	61.9
Male, %	50.9	50.8	51.7	52.9	54.0	54.2	54.9	55.6	55.7	56.4
Male incidence rate (per 100,000 person-year)										
Crude rate	237.1	239.8	238.3	244.1	253.8	261.4	270.2	265.0	274.0	271.3
Age-standardized rate	238.0	232.3	223.1	219.7	221.0	220.6	219.8	209.2	209.4	201.0
Female										
Populations	24,505,252	24,728,257	24,898,375	25,049,559	25,201,926	25,371,756	25,522,008	25,668,348	25,822,173	25,968,809
Incidence number	56,496	57,938	55,975	54,829	54,838	56,408	56,812	54,506	56,569	54,663
Mean age, year	66.8	67.1	67.4	67.5	67.6	67.8	68.0	68.2	68.6	69.2
Female, %	49.1	49.2	48.3	47.1	46.0	45.8	45.1	44.4	44.3	43.6
Female incidence rate (per 100,000 person-year)										
Crude rate	230.5	234.3	224.8	218.9	217.6	222.3	222.6	212.3	219.1	210.5
Age-standardized rate	182.4	179.0	165.8	156.0	150.3	148.9	143.9	133.3	133.1	123.4

HD—heart disease.

**Table 2 ijerph-17-08451-t002:** Annual incident numbers and incidence rates of HD Subtypes in Korea, 2006–2015.

Variables	2006	2007	2008	2009	2010	2011	2012	2013	2014	2015	*p* for Trend
IHD (I20–I25)											
Total, number	80,040	82,565	80,597	81,182	83,199	84,165	86,280	83,150	86,025	82,296	0.1178
Mean age, year	62.8	63.2	63.5	63.7	63.8	64.1	64.2	64.4	64.6	64.9	
Crude rate	162.6	166.2	161.2	161.4	164.5	165.3	168.6	161.6	166.2	158.2	
Age-standardized rate	143.9	141.9	132.9	128.3	126.5	123.1	121.1	112.8	112.5	103.5	
Arrhythmia (I47–I49)											
Total, number	12,156	13,476	14,177	15,607	16,485	18,258	19,747	20,424	21,755	22,325	<0.0001
Mean age, year	59.7	60.6	61.2	61.1	61.1	61.8	62.4	62.6	63.2	63.9	
Crude rate	24.7	27.1	28.4	31.0	32.6	35.9	38.6	39.7	42.0	42.9	
Age-standardized rate	22.6	24.1	24.5	26.0	26.5	28.4	29.5	29.6	30.4	30.1	
HF (I50)											
Total, number	13,351	14,023	14,637	14,459	15,080	16,060	16,394	16,432	17,072	18,123	<0.0001
Mean age, year	73.3	73.8	74.0	74.5	74.5	74.9	74.9	75.3	75.4	76.1	
Crude rate	27.1	28.2	29.3	28.8	29.8	31.5	32.0	31.9	33.0	34.8	
Age-standardized rate	25.6	25.5	25.2	23.6	23.4	23.7	22.9	21.8	21.5	21.7	
HD other (I00-I09, I30-I33, I39, I40-I46, I51)											
Total, number	10,007	10,653	10,939	11,234	11,938	13,006	13,503	13,609	14,907	15,609	<0.0001
Mean age, year	60.2	61.1	61.5	61.7	62.6	62.6	63.3	63.8	64.4	65.4	
Crude rate	20.3	21.4	21.9	22.3	23.6	25.5	26.4	26.5	28.8	30.0	
Age-standardized rate	18.7	19.2	19.0	18.8	19.2	20.3	20.2	19.7	20.7	20.9	
HHD (I11, I13)											
Total, number	8350	8544	7996	7142	6800	7115	6979	6347	6161	6317	<0.0001
Mean age, year	72.1	73.3	73.7	73.6	74.0	73.7	73.6	73.2	73.0	74.1	
Crude rate	17.0	17.2	16.0	14.2	13.4	14.0	13.6	12.3	11.9	12.1	
Age-standardized rate	15.9	15.4	13.7	11.6	10.5	10.5	9.8	8.5	7.9	7.7	
Pulmonary HD (I26–I28)											
Total, number	1799	2277	2605	2822	3277	3746	3727	3966	4253	4471	<0.0001
Mean age, year	62.8	63.6	63.8	65.3	66.2	65.7	66.6	67.5	68.0	68.3	
Crude rate	3.7	4.6	5.2	5.6	6.5	7.4	7.3	7.7	8.2	8.6	
Age-standardized rate	3.4	4.1	4.5	4.7	5.2	5.8	5.5	5.6	5.8	5.9	
Valvular HD (I34–I38)											
Total, number	3122	3187	3509	3587	3597	3749	3901	3918	4048	4437	<0.0001
Mean age, year	60.4	61.2	62.0	63.0	63.5	65.0	65.3	65.6	67.4	67.2	
Crude rate	6.3	6.4	7.0	7.1	7.1	7.4	7.6	7.6	7.8	8.5	
Age-standardized rate	5.8	5.7	6.0	5.9	5.7	5.7	5.7	5.6	5.4	5.8	

HD—heart disease; IHD—ischemic heart disease; HF—heart failure; HHD—hypertensive heart disease.

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
