# Peer review of "Trends in Incidence and Case Fatality Rates of Heart Disease and Its Subtypes in Korea, 2006–2015"

_ijerph, 2020, doi:10.3390/ijerph17228451_

Round 1

Reviewer 1 Report

The manuscript seems to cover an interesting topic as it examines the changing trends of the incidence and case-fatality rates of cardiovascular diseases Korea (2006 through 2015),the presented data are wide, and methodology appears appreciate. Some comments/questions can include as follows:

  1. I would suggest not to duplicate the same information in tables and in the text in the Results section
  2. The authors could present any results of statistical analysis, e.g. according to changes in trends for particular CV diseases, differences in relation to the age/sex of patients, etc. (Results)
  3. The authors state that they did not examine this, but what risk factors might be responsible for such trends over the last 10 years? (Discussion)
  4. What action should be taken to reduce demonstrated rates for CVD incidences? (Discussion)
  5. How the authors could compare the obtained results with other countries? (Discussion)

Author Response

Response to Reviewer 1 Comments

Point 1: I would suggest not to duplicate the same information in tables and in the text in the Results section.

Response 1: I deleted the same information in tables and in the text in the Results section.

Point 2: The authors could present any results of statistical analysis, e.g. according to changes in trends for particular CV diseases, differences in relation to the age/sex of patients, etc. (Results)

Response 2: I added statistical analysis (Cochran-Armitage trend test) according to the changes in trends for incidence of HD and its subtypes.

Point 3: The authors state that they did not examine this, but what risk factors might be responsible for such trends over the last 10 years? (Discussion)

Response 3: I added some studies results about risk factor (smoking, hypertension, diabetes hypercholesterolemia, obesity in Korean National Health Nutrition Examination Survey) of HD and mentioned these trends of risk factors could not explain enough decreasing pattern of age-standardized incidence patterns.

Point 4: What action should be taken to reduce demonstrated rates for CVD incidences? (Discussion)

Response 4: I mentioned ‘the first comprehensive plan for cardiovascular disease’ in 2018 and representative monitoring system and monitoring with same way continuously are important.

Point 5: How the authors could compare the obtained results with other countries? (Discussion)

Response 5: This study included incidence and case-fatality rates by sex and age-groups about a lot of subtypes of HD. So it’s not easy to compare with other counties result in a manuscript. Comparing and discussion about incidence and case-fatality rates with other counties required further studies in depth.

Reviewer 2 Report

In this manuscript titled, " Trends in Incidence and Case-Fatality Rates of Heart Disease and its Subtypes in Korea, 2006-2015", Su Ra Seo et al., authors aimed to examine the national trends in incidence and case-fatality rates of overall HD and its subtypes in Korea between 2006 and 2015. Overall, the manuscript is written clearly.  However, the manuscript appears preliminary.

  1. The abstract needs to be rewritten. It should be included the following in this specific order: what is known; what is unknown; what is the aim of the study; main information about the methods, results (no more than 2-3 sentences); conclusions (at most 2 sentences).
  2. Why did the authors select the years from 2006 through 2015 to do this study? As mentioned in reference 10, HD carries death rate in 2018 is available in Korean Statistical Information System.
  3. Authors need to supply the detailed methods that they used to do analysis in this study.

Author Response

Response to Reviewer 2 Comments

Point 1: The abstract needs to be rewritten. It should be included the following in this specific order: what is known; what is unknown; what is the aim of the study; main information about the methods, results (no more than 2-3 sentences); conclusions (at most 2 sentences).

Response 1: I revised the abstract.

Point 2: Why did the authors select the years from 2006 through 2015 to do this study? As mentioned in reference 10, HD carries death rate in 2018 is available in Korean Statistical Information System.

Response 2: I applied the Institutional Review Boards of Seoul National University on September in 2018 and approved on January in 2019. So, the available recent data in Korean Statistical Information System was in 2017.

Point 3: Authors need to supply the detailed methods that they used to do analysis in this study.

Response 3: I added the populations in Korea by sex and age-groups in supplementary table 1 and revised the methods.

Reviewer 3 Report

The first problem to be elucidated is the real accuracy of diagnostic procedures to establish the cause of death.  The second one is the statistical analysis that could be improved.

The discussion needs a more deep argumentation to understand the results

Author Response

Response to Reviewer 3 Comments

Point 1: The first problem to be elucidated is the real accuracy of diagnostic procedures to establish the cause of death.

Response 1: I added accuracy rate of statistics on the cause of death (91.9%) in manuscript and reference (Won, T.Y.; Kang, B.S.; Im, T.H.; Choi, H.J. The Study of Accuracy of Death Statistics. J Korean Soc Emerg Med. 2007, 18, 256-262. Available online: https://www.koreamed.org/SearchBasic.php?RID=2331664. (accessed on 9 November 2020)).

Point 2: The second one is the statistical analysis that could be improved.

Response 2: I added statistical analysis (Cochran-Armitage trend test) according to the changes in trends for incidence of HD and its subtypes.

Point 3: The discussion needs a more deep argumentation to understand the results

Response 3: I added some studies results about risk factor (smoking, hypertension, diabetes hypercholesterolemia, obesity in Korean National Health Nutrition Examination Survey) of HD and recent public health policy for management of HD in Korea.

Reviewer 4 Report

The authors have performed this study very well based on the available data. I have few suggestions to include:

  • The author should provide data for the total population, categorical population based on age and sex for each year to make this study more justifiable.
  • The HD subtype incidence analysis should also include frequency/total no of first incidence in each group and re-current incidence, as the first incidence of any HD increases the risk of concurrent incidence of increased.
  • It should also include the effect of sex on case-fatality rates.

Author Response

Response to Reviewer 4 Comments

Point 1: The author should provide data for the total population, categorical population based on age and sex for each year to make this study more justifiable.

Response 1: I added the information for the total population, categorical population based on age and sex for each year in supplementary table S1.

Point 2: The HD subtype incidence analysis should also include frequency/total no of first incidence in each group and re-current incidence, as the first incidence of any HD increases the risk of concurrent incidence of increased.

Response 2: The aims of this study were to provide an overview about incidence and case-fatality patterns for 10 years of HD and its subtypes, so further studies are required for analysis (including re-current incidence) in depth.

Point 3: It should also include the effect of sex on case-fatality rates.

Response 3: I added the information of case-fatality rates by sex for each year in supplementary table S10-11.

Round 2

Reviewer 3 Report

The paper is suitable for its publication in the last version